# Control Violence Begins in Adolescent Dating: A Research from Students’ Perception

**DOI:** 10.3390/ijerph19158974

**Published:** 2022-07-23

**Authors:** Josefina Lozano-Martínez, Irina Sherezade Castillo-Reche, Francisco José Morales-Yago, Francisco Javier Ibáñez-López

**Affiliations:** 1Department of Didactics and Scholar Organization, Campus de Espinardo, Universidad de Murcia, 30100 Murcia, Spain; lozanoma@um.es; 2Department of Sociology, Campus de Espinardo, Universidad de Murcia, 30100 Murcia, Spain; fj.morales@um.es; 3Department of Research and Diagnostic Methods in Education, Campus de Espinardo, Universidad de Murcia, 30100 Murcia, Spain; fjil@um.es

**Keywords:** teenagers, control, mistreatment, violence, bullying, online abuse

## Abstract

The purpose of this study was to analyze the prevalence of controlling violence experienced by adolescents in the Region of Murcia, as well as to analyze the patterns and sociodemographic variables involved such as sex, age, religious beliefs, sexual orientation, and country of origin of the families with the consequent cultural background provided. Using a sample of 454 secondary and high school students who completed a survey, the results revealed that 29.96% of the respondents were perpetrators (exerted violence) and 35.68% were victims of at least one dating abuse behavior. Significant differences were found in the occurrence of abuse based on family background, age, and religion. Finally, the results revealed that there were no significant differences in the victimization or perpetration of violence in relation to sex, but the older the victim, the less control exercised in cybernetic media, and the greater the control of the other in relation to family origin, where those from Latin American and African countries showed a greater propensity to control their partners than those of Spanish origin.

## 1. Introduction

People in general, and especially in the period of adolescence, need to create solid and permanent interpersonal bonds that serve as an adaptive and survival environment [1]. Sometimes this search gets too intense, affecting people and their relationships in a destructive way [2]. Situations of great emotional dependence appear that generate anxiety, insecurities, and fear of loneliness. As well as the emptiness of not being reciprocated by the other as one expects or desires; we also find issues of idealization of the other and blinding that prevents seeing the abuse or deprivation of freedom that may arise and are not easy to detect [3].

Situations of abuse can begin in courtship, reproducing models of inequality in their relationships and sometimes responding to sexist stereotypes in their behavior, which are nuanced by the conditions of the cultural context, where, in many cases, traditional and sexist gender stereotypes are reinforced, which undoubtedly facilitate the emergence of violence in the couple [4,5]. If we review the body of literature and evaluation studies on teen dating violence or abuse [6,7,8,9], we find that there is no uniform definition of such violence. However, there is a clear consensus that, in teen couples, adolescent violence resembles adult domestic violence as a pattern of abusive behavior used to control the other person [10]. Hence, dating violence can be conceived as a set of repressive behaviors, used by adolescents to control their partners, and entailing certain behaviors that may include abuse, permanent emotional control, intimidation, blaming, possessiveness, degradation, humiliation, minimization of feelings, and physical and sexual aggression [4,11,12].

The latest report from the National Institute of Statistics [13] regarding data from 2021, indicates that violence is growing the most in those under 18 years of age. Specifically, adolescents are the age group in which the number of reported cases has increased the most compared to 2020: 70.8%. And among adolescents, the number of victims has increased the most: 28.6%. This problem has been increasing in recent years, since the observation of the annual statistics of the National Institute of Statistics identified that from 1 January 2011, to 31 December 2021. There have been 323,063 confirmed victims of gender violence, with 2019 being the year of highest incidence with a total of 31,911 cases. The year of highest rise was 2018, where cases increased with respect to the previous year in a total of 2178 more cases, which means 7.31%. But the most striking fact, in relation to our study, is that some of these complaints are from girls under 18 years; specifically in the year 2021, were 0.9%, which means an absolute number of 287. Legal resources have not proved sufficient effectiveness to stop this type of behavior, therefore, there is a lack of protection that is getting even worse with the passage of time [14]. In most cases, attacked people keep this situation secret and find it difficult to seek the support of their relatives and authorities who can intervene This knowledge is necessary in the direction of educational centers or even a complaint to the police. Fear of rejection or lack of truthfulness is one of the causes of this lack of reporting. Therefore, it is obvious that the role of school can be decisive in the denunciation of these behaviors, hence the need to have an observant, proactive teacher, able to empathize with the victims and help them to get out of these denigrating and unfair situations. We also believe that the dissemination and development of important material created to work on the prevention and resolution of these problems can be of great help in the centers [15,16,17].

All this matter is urgent to the extent that there is a setback with respect to youth awareness of machismo and violence, naturalizing it [18,19,20]. According to the barometer on youth and gender of the Fundación de Ayuda contra la Drogadicción [21]. Almost half of young men normalize gender-based violence, so that 20% of men aged 15–29 years consider that male violence does not exist and that it is just an “ideological invention” [22].

Other studies focused on violence in adolescent couples [4,23] show that adolescents do not perceive certain behaviors as signs of violence, interpreting them from the ideas of romantic love as signs of affection. These attitudes indicate confusion and lack of discernment when young people do not determine the limits between normalized treatment and situations of violence. The type of violence detected is bidirectional [24], used as a method to resolve conflicts in the relationship or because of control and jealousy behaviors that arises regardless of the sex of the subjects.

Even sometimes, the control exercised through virtual environments on the partner is not perceived as violent behavior either [25]. In relation to this we find that “checking their partner’s cell phone” is the most widespread behavior among those adolescents (13.2%) who acknowledge having exerted some act of partner violence [26], other episodes of control are contemplated, but with minor percentages such as “saying who they can or cannot talk to” (6.1%), “insulting or humiliating” (4.2%) and “controlling everything they do” (4.1%). It is noteworthy, that physical violence, “hitting”, is present in 3% of the cases and sexual violence in 3.8% [4]. It is interesting to mention that the most frequent subtype of aggression among adolescent couples is verbal-emotional violence, regardless of sex [27]. Boys acknowledge having committed more relational and sexual violence, while girls report more physical and verbal-emotional violence [8,28]. If we consider the relationship between religion and sexist attitudes, several studies conducted in different contexts indicate that the existence of religious beliefs contributes to an increase in sexist attitudes [3,29,30].

In terms of age, we found contradictory research, with no consensus on the issue. According to some authors, the older the age, the lower the prevalence of violence [31,32], while others point out that it increases with age [33], and some indicate that there are no differences based on the age of young people [22].

On the other hand, culture can affect how affective dating relationships are established among younger people [4,5], as cultural differences often influence what is understood as a dating relationship and how behaviors are interpreted or naturalized [2,18,19,20,25], influencing normalization and legitimization [34]. Therefore, the conditions of the cultural context and the origin of the family may affect the emergence and sustenance of controlling behaviors.

In relation to sexual orientation, the studies reviewed do not identify that it influences intimate partner violence [35]. One of the main elements in force in emotional control that led to abuse is derived from the misuse of new technologies. The widespread use of cell phones and the different devices available to adolescents are causing serious situations [36]. Among the different forms of control exercised through ICTs [37,38], we find: making threatening messages, dissemination of compromising images, spreading rumors and personal discrediting, use of passwords, location control of the partner or making constant messages and calls, as well as checking the cell phone of the other to see with whom he or she relates [39]. Situations of cyber-violence and harassment in networks are a current problem, and far from being mitigated, they continue to grow without effective solutions. Constant media campaigns and the work developed in educational centers is very limited and no great results have been experienced [40].

The justification of the topic comes from the analysis of reality. We have verified that before the appearance of interpersonal violence there is an important set of laws that try to prevent the emergence of this violence among adolescents and despite this. This fact has increased considerably its intensity and assiduity and has become a serious problem in the current 21st century [41], and no solutions have been found yet to attack it forcefully [2] both from the field of physical abuse and through the use of ICT [7,8,11,42,43], or other technological means [9,44].

The purpose of this article is to present the results, in the Region of Murcia, of a major R&D&I research project developed in three autonomous communities of Spain in different educational centers in Andalusia, Murcia and Castile-La Mancha. The general purpose of this project is to evaluate violence in adolescent couples (VAC) and to make improvement proposals that contribute to the prevention and mitigation of this social scourge. In the present case, the results refer to 454 adolescents from different secondary schools in Murcia.

Therefore, the general goal of the present study is to assess the prevalence of controlling violence experienced by adolescents in the Region of Murcia, and analyzing the patterns and sociodemographic variables involved, such as sex, age, religious beliefs, sexual orientation, and country of origin of the families with the consequent cultural baggage provided.

## 2. Method

### 2.1. Data Analysis

The data collected were processed and analyzed with the free software statistical package R [45]. To search for significant differences in the questions according to sociodemographic variables, nonparametric tests were applied, since these are the most robust tests for detecting significant differences in ordinal data [46]. Specifically, the Mann-Whitney U test was used for independent variables with two levels and the Kruskal-Wallis H was used for independent variables with more than two levels (*p*-value less than 0.05 and significance level α = 0.05 were taken). The effect size was calculated using the Cohens’ d and the eta squared and the post-hoc was performed with the Pairwise Wilcoxon Rank Sum Test with Bonferroni correction. These nonparametric tests act on the median of the data, although for a better understanding of the data analyzed, the mean and standard deviation of the data are also presented in the descriptive statistics tables.

### 2.2. Participants

A total of 454 adolescents aged between 13 and 16 years participated in this research work completing the survey administered, according to the distribution shown in Table 1. Regarding sex, a total of 253 participants were girls (55.73%) and 191 boys (42.07%), while 10 people chose other options (2.2%). For statistical inference on this variable, we tried to ensure that data were balanced to guarantee the validity of the results, so we uncategorized participants who had selected “other options”.

When asked if they professed any religion and, if so, which one, the majority, 60.13% (273 participants) said they were Catholic, 25.33% (115 participants) said they were atheist or agnostic, 7.9% (36 participants) said they were Muslim and, finally, 6.69% said they professed other religions.

Figure 1 shows the country of origin of the participants’ parents. As the figure shows, most participants’ parents were from Spain (69.4%). Again, for inference purposes, options were recoded into three: both parents from Spain (315 participants, 69.4%), both parents from outside of Spain (92 participants, 20.2%) and one parent from Spain and the other from outside of Spain (47 participants, 10.4%).

Finally, Table 2 shows the sexual orientation declared by participants at the time of completing the survey. The majority, more than 82% of those surveyed, said they were heterosexual. Again, for inference purposes, this variable was recoded into two levels: heterosexual (374 participants, 82.4%) and another option (80 participants, 17.6%).

### 2.3. Instrument

A Likert scale-type questionnaire Teen Dating Violence. Victimization and Perpetration (TDV-VP) designed and validated by Soriano-Ayala et al. [47] was administered to participants. This instrument has a total of 47 items relating to the perception of intimate partner violence divided into two dimensions, violence received, violence perpetrated, and ten more sociodemographic questions. The reliability of the questionnaire was checked again with the calculation of Cronbach’s Alpha and a value of α = 0.939 was obtained, considered to be excellent [48]. Due to the ordinal nature of the scale, the Composite Reliability and McDonald’s Omega indices were also calculated, obtaining values of 0.932, considered to be excellent [49] and 0.951, also excellent [50].

For the work presented here, Table 3 shows the 14 items of the first dimension (violence received/victimization) and their 14 equivalent items of the second dimension (violence exerted/perpetration) that were analyzed relative to control violence. Participants were asked to rate intimate partner violence with questions on a scale of 1 to 4, where 1 corresponded to “Never”, 2 to “Sometimes”, 3 to “Very often” and 4 to “Always”.

### 2.4. Procedure and Information Treatment 

Questionnaires were administered via web using the digital application Google Forms. Through the web link and before completing the questionnaire, participants were informed of the purpose of the research and of their right to withdraw from the study at any time. Previous authorization had been requested from families. It was also reported that the research had the approval of the Bioethical Committee of Human Research of the University of Almeria (code Ref.: UALBIO2020/003).

## 3. Results

### 3.1. Control Violence Received

The results obtained in the questions of this dimension are shown in Table 4. A total of 162 participants (35.68%) reported having received some type of controlling violence. Questions I1.4, I1.12, I1.13 and I1.14 stand out, with response results at the “Never” level below 90%. Thus, 11.45% admitted that at some point their partner prevented them from doing something they wanted to do with comments (I1.4), 18.50% stated that their partner got angry if they were online and did not answer (I1.12), 14.54% indicated that their partner was aware of whether they were online or connected to a social network (I1.13) and, finally, 17.84% confirmed that their partner was jealous of the comments they received on social networks (I1.14).

Regarding the search for significant differences in these questions, Table 5 shows the variables and categories in which they were identified.

Initially, with respect to the variables sex, age and sexual orientation, no significant differences were detected.

With respect to the parents’ country of origin, significant differences were detected in question I1.1, on whether the partner does not let them chat with some friends and gets angry if they do (*H*(2) = 6.16, *p*-value = 0.04, eta squared = 0.01) and in question I1.11, on whether the partner has tried to gain access to their social network account (*H*(2) = 7.14, *p*-value = 0.03, eta squared = 0.02). Performed post-hoc, for question I1.1, differences were detected between participants with both parents from Spain ($\bar{x}$ = 1.10 and median = 1) and those with parents from outside of Spain ($\bar{x}$ = 1.26 and median = 1). For question I1.11, they were detected among those whose parents were both from Spain ($\bar{x}$ = 1.10 and median = 1) and those whose parents were both from outside of Spain ($\bar{x}$ = 1.25 and median = 1). 

Finally, with respect to the religion variable, significant differences were detected in questions I1.1, on whether the partner does not let them chat with friends (*H*(5) = 9.36, *p*-value = 0.03, eta squared = 0.04) and in question I1.13, on whether the partner is aware of whether they are online or connected to social networks (*H*(5) = 8.06, *p*-value = 0.04, eta squared = 0.03). In the post-hoc, for question I1.1 differences were detected between Catholics ($\bar{x}$ = 1.16 and median = 1) and agnostics/atheists ($\bar{x}$ = 1.39 and median = 1), and for question I1.13 differences were again detected between Catholics ($\bar{x}$ = 1.31 and median = 1) and agnostics/atheists ($\bar{x}$ = 1.54 and median = 1).

### 3.2. Control Violence Exerted

Table 6 shows the results obtained in the questions referring to control violence exerted. For these questions, a total of 136 participants (29.96%) admitted having exercised some type of controlling violence over their partner. In this case, questions I2.12, I2.13 and I2.14 stand out, presenting response results at the “Never” level below 90%. Some 17.40% stated that at some point they had become angry with their partner if he/she was online and did not answer (I2.12), 12.78% indicated that they were aware of whether their partner was online or connected to a social network (I2.13) and, finally, again 12.78% confirmed that they felt jealous after reviewing the comments received by their partner on social networks (I2.14).

Regarding the search for significant differences in these questions, Table 7 shows the variables and categories in which they were identified. 

For these questions, initially, no significant differences were detected with respect to the variable sexual orientation. 

Regarding sex, differences were detected in question I2.8, on whether they have spied on their partner’s things (*W* = 25,974, *p*-value = 0.00, Cohens’ d = 0.29) as there were more affirmatives responses from women, in question I2.11, on whether they had tried to gain access to their partner’s social network account (*W* = 25,034, *p*-value = 0.04, Co-hens’ d = 0.22) with a higher number of responses from women, and in question I2.12, on whether they get angry if they see that their partner is online and does not answer right away (*W* = 26,136, *p*-value = 0.03, Cohens’ d = 0.23) with the highest number of affirmatives responses from women again.

With respect to age, significant differences were detected in question I2.1, on whether he/she does not let the partner chat with friends and gets angry (*H*(3) = 10.75, *p*-value = 0.01, eta squared = 0.04), in question I2.8, on whether they have spied on their partner’s things (*H*(3) = 18.76, *p*-value = 0.00, eta squared = 0.07) and on question I2.11, on whether they had tried to gain access to their partner’s social network account (*H*(3) = 9.21, *p*-value = 0.03, eta squared = 0.04). Performed post-hoc, for question I2.1, differences were detected between participants aged 14 years ($\bar{x}$ = 1.22 and median = 1) and those aged 16 or older ($\bar{x}$ = 1.03 and median = 1). For question I2.8, they were detected among those who were 13 years old or younger ($\bar{x}$ = 1.44 and median = 1) and those who were 16 years old or older ($\bar{x}$ = 1.06 and median = 1). Finally, in question I2.11, they were again detected among those who were 13 years old or younger ($\bar{x}$ = 1.52 and median = 1) and those who were 16 years old or older ($\bar{x}$ = 1.44 and median = 1).

Finally, with regard to the religion variable, significant differences were detected in question I2.2, on whether they had deleted or blocked friends of their partners in their social media or their mobile phone so that they could not have any contact with them (*H(3)* = 8.40, *p*-value = 0.04, eta-squared = 0.03). At the post-hoc, differences were detected between agnostics/atheists (and median = 1) and Muslims (and median = 1).

## 4. Discussion

Through the results, we can observe that more than a third of the respondents claim to have received controlling violence from their partner, and a little less than that third of the sample, reports having exercised, on some occasion, some type of controlling violence, especially when this is mainly carried out through mobile devices. These results confirm those obtained by various authors who have investigated this prevalence, placing it in similar ranges, although slightly higher than those obtained in this research [7,8,9,42,44,51,52].

After analyzing whether sex, age, parents’ country of origin and religion influence young people’s perception of exerted or perceived controlling violence, significant differences were observed considering some of these factors, which will be analyzed and discussed in greater detail below.

In relation to sex, no significant differences were observed in victimization, but they were in perpetration of violence. These results confirm the findings of other re-search works, where girls exercise more control violence [8,19,22,27,43,53]. The results differ with those obtained by other authors, who found that these types of behaviors were more unequally used, finding greater victimization in girls [36,51,54,55] and with other studies that concluded that such violence was used by both boys and girls in a similar way [9,24,56,57,58].

In relation to the age factor, differences were identified in the violence exerted but not in the violence received. This differs from authors [22] who do not identify differences in violence perpetrated or exerted according to age. It has been observed that behaviors such as limiting who the partner chats with, spying or trying to access the private account of a social network are less exercised by those who are older. Therefore, the results indicate that “the older the age, the lower the prevalence of control violence”, which confirms the findings of other authors [31,32] and differs from the results obtained by authors [33] who identified that both victimization and perpetration of dating violence tended to increase slightly with age.

Regarding culture, in the present research, similarly to other authors [58], they found subtle differences, identifying some differences in relation to the country of origin. In this study, no differences were found in relation to the country of origin, but some ethereal differences were found in relation to the family’s country of origin. Thus, higher rates of controlling violence were observed in those whose parents come foreign country compared to those whose parents are from Spain in the questions: “my partner does not let me chat with some friends and gets angry if I do“ and “I have spied on my partner’s things (phone, emails, social networks, etc.)”. The observation of higher rates of violence in the sample from foreign countries joins the conclusions observed in studies by authors [59] in which they concluded higher rates of violence in all items in the sample from Peru than in the sample from Spain.

Also, differences have been observed between those with both parents from foreign countries and those from Spain, identifying a greater attempt to access social network passwords in those whose families come from abroad.

It appears that ethnicity has an inconsistent relationship with both violence victimization and perpetration [58,60]. The relationship between ethnic minorities and dating violence may be linked by other sociocultural factors, such as socioeconomic status, place of residence, or family structure.

In relation to religion and the realization or reception of controlling behaviors, significant differences were also observed in contrast to another research work [19]. They identified that those who belong to the Catholic religion indicate receiving behaviors such as their partner not letting them chat with friends or being aware of whether they are connected to a lesser extent than those who are agnostic. So, this study joins others [61,62] that support that religion can act as a protective factor against violent behaviors [63], by becoming a coping strategy for problems in young couples [64].

However, according to the present study, religion does not always act in favor of violence prevention, if we observe the data obtained in the perpetration of this, we identify that those who belong to the Muslim religion recognize that they eliminate or block friends of their partners to prevent them from having contact with them to a greater extent than those who do not process any religion. These findings are in line with several studies that identify that sexist attitude increases as religiosity does [3,29,30,65].

It has been evidenced that the violence of control exerted by adolescent partners crosses borders but is affected by the cultural patterns instilled by lived in families, hence it is more prevalent in those cultures, such as African or South American, where behaviors of control over other family members are seen with “certain normality”. If we add to this the influence of religion, control becomes worse, as has been proven by the data of the present research, we continue to verify that Muslim adolescents recognize that they eliminate or block friends of their partners so that they do not have contact with them to a greater extent than those who do not follow any religion.

Finally, in relation to sexual orientation, it should be noted that no differences were found. No differences were detected in the studies analyzed, according to which the sexual orientation of people in relation to partner violence has no influence [35].

## 5. Conclusions

Thus, and as a general conclusion, authors observed that new technologies are changing the way in which young people live their affective dating relationships [66], sometimes becoming tools to exercise control violence on the partner [26]. The prevention of this violence should be a focus of professional interest [58], since it is a growing problem among adolescents [67], affecting, as highlighted, one third of young population.

To this end, it is necessary to make a greater investment to prevent dating violence among adolescents, which allows for the development of actions that respond to and prevent this situation. These actions must take into account that:-Girls admit to exercise more controlling violence, as they spy on their partners, try to access their accounts on social media and get angry if their partner is online and does not respond.-Control violence is exercised to a greater extent by young adolescents. Preventive actions should therefore start during early adolescence.-Actions should consider the different cultural patterns, paying special attention to the country of origin of the families, as it has been observed that controlling violence is more prevalent in cultures from families of foreign origin.-Religion is a relevant factor in the approach to any intervention, as sometimes it may act as a protective factor, as in the case of Catholic religion, or as a predisposing factor for suffering this violence, as in the case of the Muslim religion.-No significant differences were found in the study with respect to sexual orientation and the country of origin of the participants.

A limitation of the present study was that we did not have a larger sample of participants when collecting students’ assessments during the pandemic. The fact that we were unable to access educational institution meant that the questionnaires were completed under the supervision of their guardians who, on occasions, found themselves with a small group of subjects due to the absence of those students who had contracted the disease. On the other hand, another of the main limitations was linked to the lack of time to prepare an action plan that would provide a solution to the situation described by the participants in the educational contexts under study, and that the results could be evaluated to compare the effectiveness of the plan.

In this sense, and as a projection for future research, it would be interesting to consider the importance of building proposals for improvement in educational centers, aimed at students and families, that contribute to deconstructing sexism and gender roles, that promote healthy relationships and that, at the same time, are spaces for the care and prevention of violence [17,18,25]. Another possibility lies in the construction of networks between different educational institutions that work together, even though we know that in the field of violence among adolescents, the magnitude of the problem, external influences, and the rise in ICTs as tools for socialization, pose a barrier in the prevention and awareness-raising among the young population. However, we believe that education contains a treasure that helps social transformation and that this issue is urgent, especially if we want to build societies free of violence.

## Figures and Tables

**Figure 1 ijerph-19-08974-f001:**
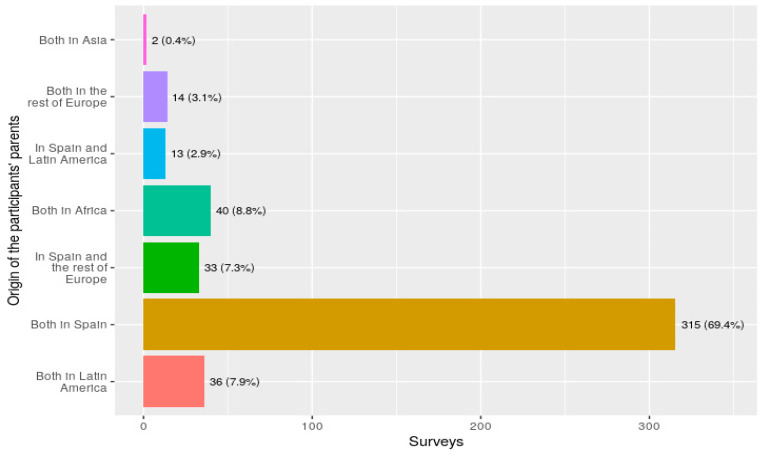
Country of origin of the parents of the participants.

**Table 1 ijerph-19-08974-t001:** Participants by age.

Age	N	%
13 years or less	66	14.54
14 years	98	21.59
15 years	195	42.95
16 years or more	95	20.93

**Table 2 ijerph-19-08974-t002:** Sexual orientation of the participants.

Sexual Orientation	N	Percentage
Bisexual	41	9.03
Heterosexual	374	82.38
Homosexual, lesbian or gay	4	0.88
Different from the previous ones	7	1.54
I do not know	28	6.17
Total	454	100.00

**Table 3 ijerph-19-08974-t003:** Issues analyzed in the two dimensions.

Violence Received	Violence Exerted
I1.1. My partner does not let me chat with some friends and gets angry if I do.	I2.1. I do not let my partner chat with some friends, and I get angry if he/she does.
I1.2. My partner has made me delete or block friends from my social networks or from my cell phone so that I do not have contact with them	I2.2. I have forced my partner to delete or block his/her friends from his or her social networks or cell phone so that he/she does not have contact with them.
I1.3. My partner has made me delete comments, photos, or videos of me on social networks because he/she was jealous of them	I2.3. I had comments, photos or videos of my partner deleted from social networks because they made me jealous.
I1.4. My partner has controlled or tried to stop me from doing something I wanted to do with comments	I2.4. I have controlled or tried to stop my partner from doing something he/she wanted to do with my comments
I1.5. My partner has tried to stop me from talking to or seeing my friends and/or family members	I2.5. I have tried to stop my partner from talking to or seeing his/her friends and/or family members
I1.6. My partner checks what I do and demands that I tell him/her where I have been	I2.6. I checked what my partner was doing and demanded him/her to tell me where he/she had been
I1.7. My partner asks me where I am every minute of the day	I2.7. I ask my partner where he/she is every minute of the day
I1.8. My partner has spied on my things (phone, emails, social networks…)	I2.8. I have spied on my partner’s things (phone, emails, social networks, etc.).
I1.9. My partner has checked through friends, family, and other means to see if it is true that I was where I told him/her I was	I2.9. I have checked with friends, family, and other means to see if my partner was where he/she told me he/she was
I1.10. My partner keeps an eye on everything I do	I2.10. I keep an eye on everything my partner does
I1.11. My partner has tried to gain access to my social network account.	I2.11. I have tried to gain access to my partner’s social network account
I1.12. My partner gets angry if he/she sees that I am online and I don’t answer him/her right away	I2.12. I get angry if I see that my partner is online and does not answer right away
I1.13. My partner keeps track of whether I am online on my cell phone or logged on to social networking sites	I2.13. I keep track of whether my partner is online on the cell phone or connected on social networks
I1.14. My partner gets jealous after reading messages I receive on my account or comments on my photos	I2.14. I get jealous after reading the messages my partner receives on his/her account or comments on his/her photos

**Table 4 ijerph-19-08974-t004:** Assessment of questions on violence received.

Question	M	Med	SD	%1	%2	%3	%4
I1.1	1.15	1	0.52	90.53	5.73	1.98	1.76
I1.2	1.11	1	0.43	92.95	4.19	1.98	0.88
I1.3	1.09	1	0.39	94.27	3.30	1.76	0.66
I1.4	1.17	1	0.51	88.55	7.05	3.52	0.88
I1.5	1.09	1	0.39	93.39	5.07	0.66	0.88
I1.6	1.14	1	0.49	90.97	5.95	1.54	1.54
I1.7	1.12	1	0.42	90.53	7.71	0.88	0.88
I1.8	1.15	1	0.51	90.31	6.61	1.32	1.76
I1.9	1.11	1	0.45	92.95	3.96	1.98	1.10
I1.10	1.09	1	0.38	93.61	4.85	0.66	0.88
I1.11	1.13	1	0.50	92.29	4.19	1.76	1.76
I1.12	1.31	1	0.74	81.50	9.69	5.07	3.74
I1.13	1.23	1	0.63	85.46	8.59	3.52	2.42
I1.14	1.28	1	0.68	82.16	10.35	4.63	2.86

**Table 5 ijerph-19-08974-t005:** Received control violence issues in which significant differences were detected according to sociodemographic variables.

Sociodemographic Variable	Question	*p*-Value	Statistic	Eta Squared	Post-Hoc
Sex	No significant differences were detected				
Age	No significant differences were detected				
Parents’ country	I1.1	0.04	*H*(2) = 6.16	0.01	Both from Spain ($\bar{x}$ = 1.10 and med = 1) with both from outside of Spain ($\bar{x}$ = 1.26 and med = 1)
	I1.11	0.03	*H*(2) = 7.14	0.02	Both from Spain ($\bar{x}$ = 1.10 and med = 1) with both from outside of Spain ($\bar{x}$ = 1.25 and med = 1)
Religion	I1.1	0.03	*H*(5) = 9.36	0.04	Catholics ($\bar{x}=1.16$ and med = 1) and agnostics/atheists ($\bar{x}=1.39$ and med = 1)
	I1.13	0.04	*H*(5) = 8.06	0.3	Catholics ($\bar{x}=1.31$ and med = 1) and agnostics/atheists ($\bar{x}=1.54$ and med = 1)
Sexual orientation	No significant differences were detected				

**Table 6 ijerph-19-08974-t006:** Assessment of exerted violence issues.

Question	M	Med	SD	%1	%2	%3	%4
I2.1	1.06	1	0.29	95.81	3.08	0.88	0.29
I2.2	1.04	1	0.28	97.36	1.54	0.66	0.28
I2.3	1.03	1	0.20	97.14	2.42	0.44	0.20
I2.4	1.04	1	0.21	96.26	3.52	0.22	0.21
I2.5	1.02	1	0.19	98.68	0.88	0.22	0.19
I2.6	1.04	1	0.23	96.70	2.64	0.66	0.23
I2.7	1.06	1	0.28	94.49	4.63	0.88	0.28
I2.8	1.11	1	0.38	92.07	5.29	2.64	0.38
I2.9	1.07	1	0.32	93.83	5.07	0.88	0.32
I2.10	1.05	1	0.31	96.70	1.76	1.10	0.31
I2.11	1.06	1	0.33	96.48	1.98	0.88	0.33
I2.12	1.26	1	0.62	82.60	10.79	5.07	0.62
I2.13	1.17	1	0.50	87.22	9.69	1.98	0.50
I2.14	1.18	1	0.52	87.22	8.81	2.86	0.52

**Table 7 ijerph-19-08974-t007:** Control violence issues in which significant differences were detected according to sociodemographic variables.

Sociodemographic Variable	Question	*p*-Value	Statistic	d Cohen/Eta Squared	Post-Hoc
Sex	I2.8	0.00	*W* = 25,974	0.29	Male ($\bar{x}$ = 1.04 and med = 1) with female ($\bar{x}$ = 1.15 and med = 1)
	I2.11	0.04	*W* = 25,034	0.22	Male ($\bar{x}$ = 1.02 and med = 1) with female ($\bar{x}$ = 1.09 and med = 1)
	I2.12	0.03	*W* = 26,136	0.23	Male ($\bar{x}$ = 1.18 and med = 1) with female ($\bar{x}$ = 1.32 and med = 1)
Age	I2.1	0.01	*H*(3) = 0.75	0.04	14 years ($\bar{x}=1.22$ and med = 1) with 16 years or more ($\bar{x}=1.03$ and med = 1)
	I2.8	0.00	*H*(3) = 8.76	0.07	13 years or less ($\bar{x}=1.44$ and med = 1) with 16 years or more ($\bar{x}=1.06$ and med = 1)
	I2.11	0.03	*H*(3) = 9.21	0.04	13 years or less ($\bar{x}=1.52$ and med = 1) with 16 years or more ($\bar{x}=1.44$ and med = 1)
Parents’ country	I2.8	0.02	*H*(2) = 7.54	0.02	Both from Spain ($\bar{x}=1.08$ and med = 1) with both from outside of Spain ($\bar{x}=1.21$ and med = 1)
Religion	I2.2	0.04	*H*(3) = 8.40	0.03	Agnostics/atheists ($\bar{x}=1.01$ and med = 1) with Muslims ($\bar{x}=1.25$ and med = 1)
Sexual orientation	No significant differences were detected				

## Data Availability

The data presented in this study are available on request from the corresponding author.

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
