# Peer review of "Control Violence Begins in Adolescent Dating: A Research from Students’ Perception"

_ijerph, 2022, doi:10.3390/ijerph19158974_

Round 1
Reviewer 1 Report
Dear authors.
Thank you very much for your valuable work. I think it is a very relevant topic, and will help to enhance the understanding of this complex phenomenon.
My suggestions are related to your conclusions. I would like to see a wider report of the limitations of your research. It would be interesting for readers to get this information. Besides, in the conclusion you mention that future research direction may be highlighted, and I would like to see wider information about this.
Thank you very much.
Author Response
Thank you for your contributions. They have undoubtedly contributed to the rigor and quality of the work and have helped for future contributions.
In the document have incorporated to the limitations at the end of the conclusions, together with the projection.

Reviewer 2 Report
Review is uploded as a pdf. file

Author Response
A continuación se responderá a cada una de las preguntas planteadas:
El artículo trata un tema importante y, desde hace mucho tiempo, actual. La lista de referencias contiene 62 fuentes de literatura reciente
Sugiero los siguientes cambios/mejoras:
- línea 16 - no es necesario escribir "..., el área en estudio, ..."; Recomiendo borrar esto, lo mismo se aplica a la línea 126
Eliminado de ambos
- después del texto que termina en la línea 50 - Recomiendo continuar con los datos epidemiológicos (texto que comienza en la línea 65, omitiendo el párrafo de la línea 51 a la 57) y revisar los datos epidemiológicos obtenidos en el proyecto; antes de describir los datos del proyecto, agregue el texto que ahora está en las líneas 51-57.
reorganizado
- la oración (líneas 58-63) es demasiado larga y por lo tanto incomprensible; recomiendo reformular
Reformulado
- a lo largo de la parte introductoria del texto, se ha dejado un punto después de los paréntesis con referencias en varios lugares (líneas 63, 90, 117)
Revisado e implementado en todo el documento.
- el texto introductorio carece de investigación sobre las diferencias en la violencia en el noviazgo adolescente por edad, creencias religiosas, orientación sexual y país de origen. Esto es importante porque habla de la justificación para examinar estas diferencias en este estudio. El texto sobre estas diferencias se puede encontrar en la discusión: el texto (otros resultados de la investigación) sobre las diferencias mencionadas debe integrarse en la introducción y luego mencionarse en la discusión (p. ej., sobre el país de origen (cultura) - texto en las líneas 283 -288 debe integrarse en la introducción).
Líneas 283-288 integradas en el texto introductorio, destacando también investigaciones sobre diferencias según los factores analizados.
- capítulo 2.1 - Se recomienda que se llame Análisis de datos en lugar de Diseño y que la información sobre métodos de análisis de datos se traslade a este capítulo (líneas 177-186). La última oración (líneas 132-136) no es necesaria ya que esta información se incluye en el capítulo 2.3.
Capítulo 2.1 renombrado. Líneas 132-136 eliminadas.
- capítulo 2.4. necesita describir el procedimiento con más detalle: cómo se distribuyó el cuestionario, quién dio las instrucciones, qué información se incluyó en el formulario de consentimiento, cuánto tiempo se tardó en completar el cuestionario, cómo se aseguró el anonimato, si se ofreció apoyo en caso de los participantes de la investigación se agitaron o se sintieron incómodos mientras completaban el cuestionario (ya que la parte de las preguntas se relaciona con la victimización). Y, lo más importante, ¿hay aprobación ética para la investigación? Debe mencionarse en el documento (quién lo dio y cuándo).
Se ha descrito más detalladamente el proceso y se ha destacado la aprobación del trabajo de investigación por parte del comité de ética.
- Línea 148 - Recomiendo borrar el texto "Por otro lado" de la oración;
Remoto
- Figura 1 - se recomienda hacer una breve referencia en el texto a los datos presentados en la Figura 1
Implementado
- Tabla 2 - se recomienda hacer una breve referencia en el texto a los datos presentados en la Tabla 2
Implementado
- línea 192 - se recomienda no comenzar la oración con un número
Se ha añadido “así”.
- Tabla 5 y Tabla 7: todos los datos deben estar presentes, haya o no diferencias (agregue datos de género, edad y orientación sexual en la Tabla 5; agregue datos de género y padres
país en la Tabla 7)
Las tablas muestran los ítems en los que se encontraron diferencias significativas según las variables sociodemográficas. No es necesario dar los p-valores de los 14 ítems para demostrar que no existen diferencias significativas, del mismo modo que no se dan los p-valores de las demás preguntas en el resto de variables sociodemográficas, ya que esto haría que la mesa fuera demasiado larga.
- Oración en las líneas 269-271: debe quedar claro qué parte del texto se refiere a esta investigación y cuándo se menciona la investigación de otros autores. Recomiendo dividir la oración en dos oraciones, la primera de las cuales dice: "No se encontraron diferencias significativas con respecto al género ni en la victimización ni en la perpetración de la violencia". La parte que hace referencia a otra investigación, "...ya que fue utilizado tanto por niños como por niñas de manera similar [9,26,46-270 48]". debe seguir por separado
Oración dividida.
- La conclusión debería ser "más fuerte", está escrito. Basado en los datos obtenidos, especialmente en relación con las diferencias de género, edad, creencias religiosas, orientación sexual y país de origen, recomendaciones para futuras investigaciones, así como una mayor inversión en citas entre adolescentes. debe destacarse la prevención de la violencia.
Las conclusiones se han modificado teniendo en cuenta los datos obtenidos.
- línea 329 - ¿Quiénes son "ellos" en esta oración? - poco claro y necesita ser reescrito
Se ha aclarado.
- líneas 339-341: "...seguimos comprobando que los adolescentes musulmanes reconocen que eliminan o bloquean a los amigos de sus parejas para que no tengan contacto con ellas en mayor medida que los que no profesan ninguna religión". - Este texto pertenece a la discusión, no a la conclusión; no hay argumento para resaltar esta información específica en la conclusión, ni es necesario; sería más adecuada una recomendación para el desarrollo de intervenciones preventivas basadas en los hallazgos obtenidos en relación con las creencias religiosas (al igual que recomendaciones basadas en otras diferencias analizadas)
Sujeto a discusión.
- es necesario señalar las limitaciones de la investigación
Incorporado a las limitaciones al final de las conclusiones, junto con la proyección.
Gracias por sus aportaciones. Sin duda han contribuido a la rigurosidad y calidad del trabajo y han ayudado para futuras aportaciones.

Reviewer 3 Report
The study makes a descriptive analysis of a series of topics derived from the partial analysis of a test that assesses dating violence. The paper analyzes the partial test data in relation to other sociodemographic variables.
The paper addresses a topic of relevance and does so with a descriptive approach that could be useful.
I note some issues to be solved, which have to do with the validity of the study:
The statistical treatment chosen causes the conclusions to contain errors:
Differences between sample sizes are a serious problem to implement Kruskal - Wallis (Bruner et al., 2020 doi: 10.1111/insr.12418).
The interaction between variables was not considered. For example, when comparing differences in gender, sexual orientation, could age be covarying, mediating, moderating these differences? The authors should consider this and other forms of interaction.
The authors draw conclusions about the existence of violence, but their data indicate the opposite: The value "1" on their scale is equivalent to never having been a victim or having perpetrated violence. The scale used has values from 1 to 4. The mean of their data ranges from 1.09 to 1.31 for victimization. For perpetration it ranges between 1.02 and 1.26. Its percentages also show a clear tendency towards the value 1. The great majority of its items have a response rate higher than 90% for value 1 (never having received or perpetrated violence).
The scale used has 47 items. It has two dimensions. Each dimension has five factors. The control factor in the original scale is made up of seven items (14 in total). Why use only a few items of a scale instead of using the complete scale with its corresponding factors? Psychometric properties will not be the same if a fragmented scale is used (regardless of the alpha obtained by the authors).
This implies that the scope of the work is greatly reduced by making partial use of an instrument, with a basic descriptive analysis.
The above observations prevent me from recommending the work.
The wording of the paper needs to be revised. The sentences are too long. Some of them are nearly 100 words long without a period.
The analysis of reality, subjective and without data, cannot be the support or justification of a scientific work (lines 58-61). There are statements that require evidence and the authors do not show it (lines 76-82). Even from these assertions without evidence they develop suggestions (lines 82-87). The statement in lines 109-110 also require reference.
Figure 1 should be translated into English.
It specifies that informed consent was given by the individuals involved in the study. But in this case we are talking about minors. I think it should be specified more specifically how the informed consent was obtained.
"H" should be reported, not "K".
I commend the authors for further statistical analysis. To perform moderation or mediation tests and to use the entire instrument analyzed (TDV-VP). It is a valuable work, which with proper use of more data and more analysis, will become an important contribution.
Author Response
Each of the questions raised will be answered in turn below:
The study makes a descriptive analysis of a series of topics derived from the partial analysis of a test that assesses dating violence. The paper analyzes the partial test data in relation to other sociodemographic variables.
The paper addresses a topic of relevance and does so with a descriptive approach that could be useful.
I note some issues to be solved, which have to do with the validity of the study:
The statistical treatment chosen causes the conclusions to contain errors:
Differences between sample sizes are a serious problem to implement Kruskal - Wallis (Bruner et al., 2020 doi: 10.1111/insr.12418).
We are grateful for your contributions and suggestions. Undoubtedly, your proposals and observations are very pertinent, and contribute to the greater rigor and quality of our work. Thus, we have tried to balance the data by recoding the variables with a greater level of imbalance: sex, parents' country of origin and sexual orientation. With this new recoding, we have repeated the inferential analysis, obtaining in some cases the same or similar results and detecting in other cases new significant differences. Of course, the homoscedasticity of the data was previously checked to ensure the validity of the results obtained with the non-parametric tests.
The interaction between variables was not considered. For example, when comparing differences in gender, sexual orientation, could age be covarying, mediating, moderating these differences? The authors should consider this and other forms of interaction.
Initially, we had not considered going beyond a descriptive analysis and an inferential analysis after verifying the assumption of homoscedasticity of the data with respect to the sociodemographic variables necessary to use the non-parametric tests based on the median. In future studies we will address the analysis of covariance and the adjustment of regression models to establish the profile of students who exert or receive violence. Undoubtedly, this is a challenge to be faced in future studies. The aim of the present study does not go beyond a descriptive analysis of the initial situation, and this is the reason why simple inference techniques have been used.
The authors draw conclusions about the existence of violence, but their data indicate the opposite: The value "1" on their scale is equivalent to never having been a victim or having perpetrated violence. The scale used has values from 1 to 4. The mean of their data ranges from 1.09 to 1.31 for victimization. For perpetration it ranges between 1.02 and 1.26. Its percentages also show a clear tendency towards the value 1. The great majority of its items have a response rate higher than 90% for value 1 (never having received or perpetrated violence).
We believe that since this is such an important socio-emotional issue, it is necessary to address any percentage that is far from 0. The fact that there are 10-15% of people who suffer violence is already an important result. Indeed, the average does not represent this percentage of people who are suffering violence, but even so, because of the subject matter of the article, we believe that it is very important to point out these results.
The scale used has 47 items. It has two dimensions. Each dimension has five factors. The control factor in the original scale is made up of seven items (14 in total). Why use only a few items of a scale instead of using the complete scale with its corresponding factors? Psychometric properties will not be the same if a fragmented scale is used (regardless of the alpha obtained by the authors).
This implies that the scope of the work is greatly reduced by making partial use of an instrument, with a basic descriptive analysis.
The above observations prevent me from recommending the work.
The article specifies the composition of the scale and its reliability and internal consistency values. According to the subject matter of the article, the reference items have been used and the rest have been omitted (which refer to other types of violence and not to controlling violence, which is the type of violence addressed on this occasion). It would not be possible to present the results of all the dimensions and items of the scale due to its length.
The wording of the paper needs to be revised. The sentences are too long. Some of them are nearly 100 words long without a period.
The length of the sentences has been revised.
The analysis of reality, subjective and without data, cannot be the support or justification of a scientific work (lines 58-61). There are statements that require evidence and the authors do not show it (lines 76-82). Even from these assertions without evidence they develop suggestions (lines 82-87). The statement in lines 109-110 also require reference.
We have incorporated references on the referred lines.
Figure 1 should be translated into English.
It has been translated.
It specifies that informed consent was given by the individuals involved in the study. But in this case we are talking about minors. I think it should be specified more specifically how the informed consent was obtained.
It has been specified in the article.
"H" should be reported, not "K".
It has been corrected.
I commend the authors for further statistical analysis. To perform moderation or mediation tests and to use the entire instrument analyzed (TDV-VP). It is a valuable work, which with proper use of more data and more analysis, will become an important contribution.
Thank you again for your contributions. They have undoubtedly contributed to the rigor and quality of the work and have helped for future contributions.
